# Effect of Loading Direction on Deformation and Strength of Heterogeneous Paleo Clay Samples

Shaoping Huang [1,2], Yuanhai Yang [1], Henglin Xiao [1,2,*], Wenying Cao [1], Kaiming Cao [1], Ruiming Xing [1] and Yanchao Wang [3]

[1] School of Civil Architecture and Environment, Hubei University of Technology, Wuhan 430070, China; hsp@cug.edu.cn (S.H.); yangyh@hbut.edu.cn (Y.Y.); cwy@hbut.edu.cn (W.C.); ckm991122@163.com (K.C.); 102200819@hbut.edu.cn (R.X.)

[2] Key Laboratory of Intelligent Health Perception and Ecological Restoration of Rivers and Lakes, Ministry of Education, Hubei University of Technology, Wuhan 430068, China

[3] Shanxi Province Transportation Technology Research and Development Co., Ltd., Taiyuan 030032, China; wangyanchao0602@hotmail.com

\* Correspondence: xiaohenglin@hbut.edu.cn

**Abstract:** Landslides result from weak surfaces with varying rock-soil properties, posing a significant concern for engineering and accurate deformation analysis. This study investigated the macroscopic physical and mechanical properties of paleo clay specimens during triaxial compression testing, aiming to elucidate the deformation mechanisms exhibited by these specimens under varying loading directions at both the loading and unloading ends, and numerical simulation methods were carried out to simulate actual engineering scenarios. The analysis encompasses deformation patterns, stress–strain relationships, Mohr stress circles, and numerical simulation failure cloud diagrams for soil samples under different loading directions. The results showed that the loading end of heterogeneous specimens exhibited noticeable deformations. Alteration of the loading direction induced variations in the failure mode. The position and size of the deformations for the only iron-manganese clay, loading end iron-manganese clay, and loading end reticulated clay samples changed with the clay layer at the loading end of the sample. Moreover, the stress–strain curves under different loading directions were different, with strain hardening and strain softening appearing in the two loading directions, respectively. The results of this study contribute to an in-depth understanding of the impact of the loading direction on the deformation and strength of paleo clay, thereby providing a foundation for landslide prevention and control measures.

**Keywords:** loading directions; paleo clay; deformation and strength characteristics; triaxial compression test; mechanical parameters

## 1. Introduction

Paleo clay refers to a type of sticky soil with unique characteristics, including iron, manganese nodules, and kaolin stripes, which was deposited during the late Pleistocene Q3 and earlier in the Quaternary period [1,2]. It is widely distributed in central and northern Anhui Province, the mid-eastern regions of Hubei Province, and central Henan Province in China. Given its mineral composition, mainly fine clay, calcareous, and siliceous sediment [3], paleo clay has characteristics of expansibility such as water absorption expansion and water loss shrinkage [2,4]. Despite the limited inherent strength of paleo clay, which renders it insufficient to support engineering loads, its affordability and widespread availability make it an inevitable choice for use as a foundation material in building, road, and slope construction [5]. In engineering practice, paleo clay slopes are prone to deformation and disasters such as foundation instability, landslides, and roadbed settlement under external disturbance conditions such as drying–wetting cycles and dynamic traffic loads [6,7].

Therefore, it is of great significance to clarify the strength and deformation law of paleo clay caused by loading direction for studying the weak surface of multilayer soil.

Numerous studies and practical observations have consistently demonstrated that the weak surface in multilayered rock and soil systems is frequently situated at the interface or contact surface [8–10]. The mechanical properties of loading tests exhibit significant variations under different loading directions [11]. Existing research primarily focuses on stability and bearing capacity analysis through the isolation of single-layer parameters. However, this approach falls short of achieving a comprehensive understanding of slope stability and failure mechanisms. Consequently, it is imperative to acknowledge that the critical determinant influencing the stability of a slope or foundation resides in the strength parameters of the contact surface and the weak structural surface [12]. At present, this method combined with inversion calculation is usually used to achieve a comprehensive determination, resulting in inaccurate parameters for slope stability analysis and trend prediction. The properties of clay engineering are significantly influenced by internal and external factors that refer to its physical composition and human influence, respectively [13,14]. Some recent studies have revealed that internal factors are mainly determined by soil structure, material composition, and interparticle forces [15]. Paleo clay exhibits complex engineering characteristics attributed to its properties and structural arrangement, which gives rise to varying soil layers. In general, the bonded interface between dissimilar materials is often a weak link in terms of mechanical stability [16]. The distinct physical and mechanical properties (e.g., strength, expansiveness, and permeability) among multiple soil layers often lead to the deformation and failure of slopes, buildings, as well as road foundations. Extensive laboratory and field tests have been carried out to examine the basic parameters for studying the deformation and bearing capacity of paleo clay [17–19]. For instance, research focused on the failure modes of paleo clay foundation pits was based on the examination of ancient clay minerals and their structural characteristics [4]. After studying the physical and mechanical properties as well as engineering characteristics of paleo clay, the settlement calculations of paleo clay foundations could be optimized. The analysis of the causes, distribution characteristics, and engineering properties of paleo clay can provide planning and construction data for areas rich in paleo clay [20,21].

To study the disaster-causing mechanism of paleo clay engineering, a comprehensive and systematic analysis from a mechanical perspective should be conducted to obtain the mechanical parameters of the soil contact surface and explore the factors that affect the accurate acquisition of mechanical strength parameters. However, research on the influence of the loading direction on soil deformation, strength, and failure mechanisms is still in its early stages of development. Yuke Wang et al. [22] conducted single-directional vibration cyclic axial tests on remolded and intact soft clay samples and studied the deformation laws of the two states of the test soil. The results showed that under the same load, the accumulated plastic strain produced by the remolded sample was larger. Further studies have shown that loading path and direction induce significant differences in stress–strain curves and failure modes [23,24]. Intact soil has a clear shear failure surface, while remolded soil specimens exhibit drum-like deformations after tests are completed [25]. In addition, remolded soil samples generally exhibit strain hardening. The initial consolidation pressure of specimens cut horizontally was observed to be smaller than that of specimens cut vertically after conducting tests on clay samples [26]. Costanzo D. et al. [27] conducted tests on remolded samples and described the nonlinear behavior of the soil observed in the laboratory under finite-size loading paths. Su S. et al. and Whittle A. J. et al. [28] highlighted that paleo clay exhibits anisotropic characteristics as a consequence of the anisotropic consolidation stress state developed during its formation. Wang X. L. et al. and Zhuang H. et al. [29,30] conducted studies on remolded loess samples and recycled soft clay; respectively, when the magnitude and direction of the principal stress changed simultaneously, the direction of the principal stress would affect the anisotropy and non-coaxiality of the recycled soft clay, and the intermediate stress coefficient and rotation range of the principal stress would have the greatest impact on the deformation of the specimen.

Wan Y. et al. [31] studied the influence of different sizes of specimens on the permeability coefficient of compacted clay. Li Hai Peng et al. [32] analyzed the influence of vertical and horizontal loading directions on the mechanical properties of frozen powder soil, and the results showed that the compressive strength under vertical loading was 12% higher than that under parallel loading under the same conditions.

In consideration of the facts that are mentioned above, it can be seen that most of the present studies have recognized the important role of the loading direction in clay samples; however, the effects of loading on multilayer clay strength have not been paid enough attention. Although some research has been conducted from the perspective of stress direction and loading direction, the underlying mechanisms have not been fully explained. For instance, the strength parameters and deformation characteristics of two-layered paleo clay samples change when subjected to different loading directions. To accurately obtain the mechanical parameters of heterogeneous paleo clay, it is necessary to analyze the influence of the loading direction on the mechanical parameters of samples and reveal the underlying mechanisms.

Paleo clay is widely distributed in actual slope and foundation engineering and is mainly multilayer. The strength of multilayered paleo clay is susceptible to a variety of factors, such as composition, soil structure, and loading position and direction. Therefore, it is necessary to study the mechanical response and deformation of old clay under different loading directions. This study aimed to: (1) Investigate the effect of the loading direction on the strength parameters of multilayered soil. (2) Analyze the failure characteristics of the samples. (3) Reveal the underlying mechanisms, providing a basis for accurately determining the parameters of multilayered soil. (4) Obtain more accurate mechanical parameters of the soil within the sliding zone, offering both theoretical and technical support for predicting slope evolution trends and developing precise prevention and control strategies.

## 2. Materials and Methods

### 2.1. Materials

The clay samples collected from the unstable slope in Shiguling Village, Jiayu County, Hubei Province, for this experiment were typical paleo clays. The clays contained abundant iron-manganese nodules, kaolin, and clay particles, and they were divided into iron-manganese clays and reticulated clays according to the content of iron-manganese and kaolin (as shown in Figure 1). The basic physical and mechanical parameters of the paleo clays are summarized in Table 1.

In this study, tap water was mixed with iron-manganese clays and reticulated clays, the water–clay ratios (optimum moisture content) were 22.72% and 18.10%, respectively. The optimum moisture content was determined by the compaction test.

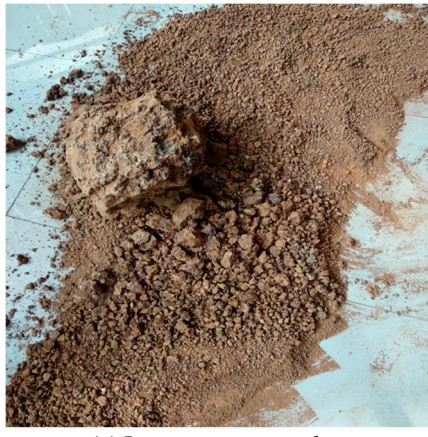 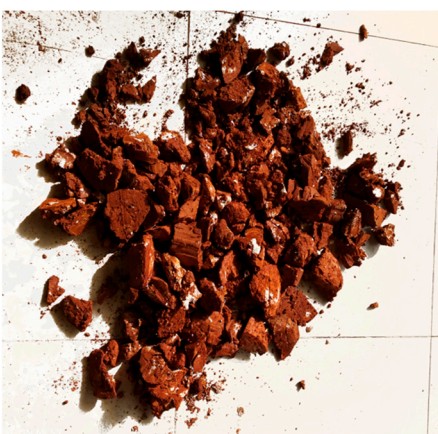

(**a**) Iron-manganese clay                          (**b**) Reticulated clay

**Figure 1.** Original soil samples.

**Table 1.** Basic physical and mechanical parameters of paleo clay.

| Name | Specific Gravity (kN·m$^{-3}$) | Liquid Limit (%) | Plastic Limit (%) | Dry Density (g·cm$^{-3}$) | Optimum Moisture Content (%) |
|---|---|---|---|---|---|
| Iron-manganese clays | 2.73 | 38.17 | 21.93 | 1.53 | 22.72 |
| Reticulated clay | 2.73 | 36.98 | 20.65 | 1.64 | 18.10 |

### *2.2. Main Test Instruments*

In this experiment, the main instruments included a strain-type triaxial apparatus (Figure 2), a TYS-50 soil compaction device (Figure 3), and a platform scale [33].

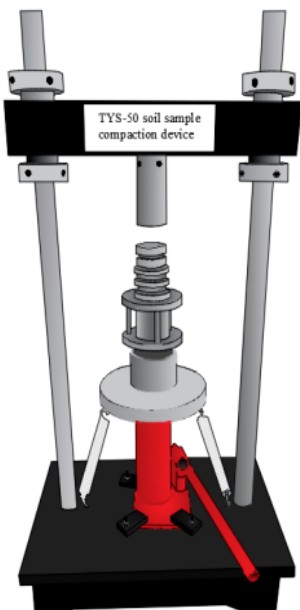

**Figure 2.** Strain-type triaxial compressor.

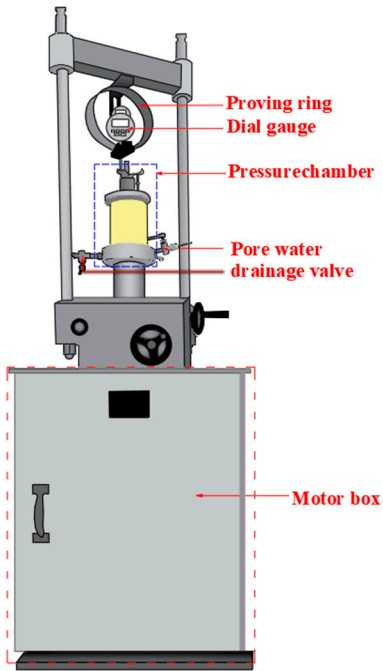

**Figure 3.** Sample-making instrument.

*2.3. Test Scheme and Process*

In order to remove the water from the clay samples, they were dried in an oven with a constant temperature of 105 °C for 24 h prior to performing the other procedures. Subsequently, the clays were crushed by wood milling and passed through a 2 mm sieve with the aim of obtaining homogeneous clay particles without impurities [34,35].

According to ASTM D2850-23 [36], the required amounts of paleo clay particles and tap water were mixed in a mixer to obtain uniform mixtures. To prepare heterogeneous paleo clay samples, they were compacted into four layers of the same height, which contained two layers of iron-manganese clays and two layers of reticulated clays. The roughness of the contact surface of each layer was increased with a planer knife to prevent slippage of each layer during the test. Hence, the whole sample preparation process was carried out until 18 standard paleo clay samples with 39.1 mm diameter and 80 mm height were obtained.

To study the influence rule of the axial loading direction on the deformation and strength of heterogeneous paleo clay samples, the test clay samples were divided into three groups (Group A, Group B, and Group C). Among them, Group A only contained iron-manganese clay samples. Group A samples were loaded in a saturator and placed in a vacuum pump for 2 h, soaked for 24 h, and then the samples were taken out and placed in an oven at 65 °C for continuous drying until constant weight of the sample was reached, that is, the dry–wet cycle was completed once, and the next cycle was carried out after cooling for 2 h, for a total of 10 cycles.

Furthermore, both Group B and Group C included clay samples with upper and lower parts respectively composed of iron-manganese clays and reticulated clays. In addition, Group B was made up of clay samples from the iron-manganese clays at the loading end during the triaxial compression tests, and Group C was made up of clay samples from the reticulated clays at the loading end during the triaxial compression tests. One sample was taken from each of the three groups of samples, and the test was repeated twice to improve the accuracy of the test results. Thus, Group A, Group B, and Group C each had six paleo clay samples.

In the triaxial compression test, it should be particularly noted that the top and bottom surfaces of the specimens needed to be trimmed to be flat enough to reduce errors caused by the stress concentration. Three groups of samples underwent triaxial compression tests with confining pressures of 100 kPa, 200 kPa, and 300 kPa. Displacement-controlled axial loading (as shown in Figure 2) was adopted, the rate of which was determined in accordance with ASTM D4767-95 [37], namely 0.5 mm/min. The test stopped when the specimens were destroyed or the axial strain reached 15%.

## 3. Results

### 3.1. Effect of Loading Direction on Deformation Characteristics of Heterogeneous Specimens

According to the test results of different loading directions, we know that the paleo clay underwent a process of gradual development from local to overall failure. The typical failure patterns of the paleo clay samples under different loading directions in the triaxial tests are given in Figure 4. As depicted in Figure 4a, when the confining pressure reaches 100 kPa, a singular prominent rupture surface is evident. This rupture surface intersects the specimen at an angle of approximately 60° relative to the vertical direction and extends through the entire specimen. During the loading process, the main fracture surface initially appears in the reticulated clay of the specimen. As the load continues to be applied, the main rupture surface continuously evolves and penetrates the specimen. Comparing the deformation failure characteristics and evolution law of the specimens under various confining pressures (100, 200, and 300 kPa), as shown in Figure 4b,c, it is observed that the number of cracks gradually decreases as the confining pressure increases, and the degree of the fracture surface is reduced. It is a well-known phenomenon that during the initial stage of increasing confining pressure, axial and radial deformations are more sensitive to low confining pressure, and the strength of

the clay will increase accordingly. Therefore, when it reaches a certain value, the soil structure will be damaged and the shear strength of the clay will be greatly reduced. As the confining pressure increases, the impact of compaction on radial deformation and strain at failure will gradually diminish [38].

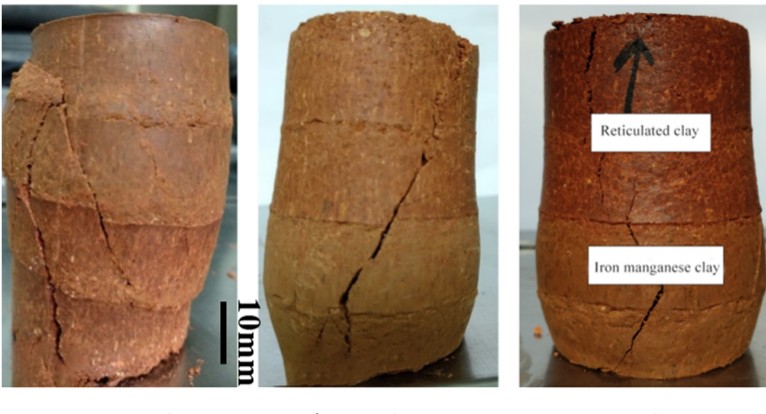

|　(**a**) 100 kPa　|　(**b**) 200 kPa　|　(**c**) 300 kPa　|

Group B: The upper reticulated clay and lower iron-manganese clay specimens.

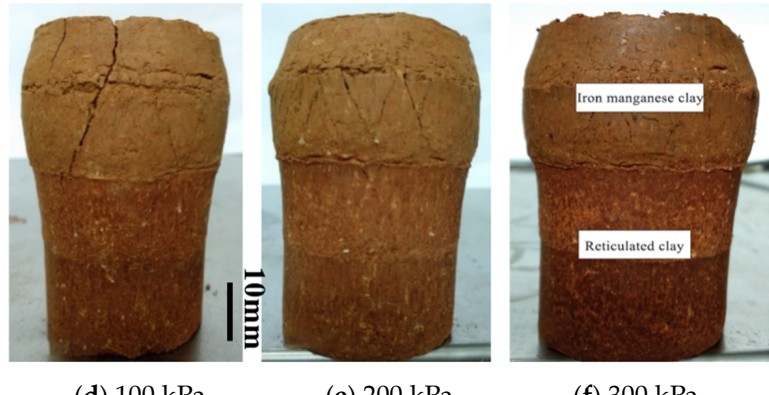

|　(**d**) 100 kPa　|　(**e**) 200 kPa　|　(**f**) 300 kPa　|

Group C: The upper iron-manganese clay and lower reticulated clay specimens.

**Figure 4.** Destruction samples of the paleo clay under different confining pressures.

　　Furthermore, increasing the confining pressure causes structural damage to the soil, leading to certain changes. The angle between the cracks and the vertical direction decreases, and the rupture surface becomes rougher. In Figure 4e, it is evident that only multiple sets of intersecting cracks are observed in the iron-manganese layered soil sample, which served as the loading end. Regardless of the confining pressure, the cracks are only found in the loading end layer, with widths spanning from 1 to 3 mm. With higher confining pressure, both the length and width of the cracks gradually decrease, consistent with the findings from specimens in Group A.

　　The findings presented above reveal that the deformation and failure mode of the specimen undergo significant changes when the loading direction changes. When the loading end consists of harder paleo clay, a primary interpenetrate surface is more likely to form during the test. However, when the loading end is composed of softer clay, deformation cracks are predominantly concentrated in the soft paleo layer, resulting in fundamental changes in the deformation failure mode. Consequently, the location of the "drum-shaped" protrusion shifts, primarily expanding within the softer paleo clay. Group C exhibits smaller crack lengths and quantities compared to Group B. The cracks and fracture surfaces in Group C are mainly concentrated at the loading end (the iron-manganese clay). This can be attributed to the fact that the softer clay in Group C is less prone to cracking due to its higher tendency to deform plastically, whereas the harder clay in Group B is

more susceptible to brittle failure. It is worth noting that the fracture surface of Group C specimens is not entirely smooth, with the upper layer of the reticulated clay appearing relatively smoother. This is due to certain bulging deformations in the lower layer of the iron- and manganese-rich clay, causing the fracture surface to widen and changing the direction of the rupture.

Figure 5 summarizes the main results of the study of the variations in specimen deformation under different confining pressures and loading directions, quantifying the deformation by measuring the variations in sample diameter. Regardless of the value of the confining pressure and frequency of dry–wet cycles, all curves exhibit a smooth bell-shaped trend, increasing to a peak value and then decreasing. The specimens show the end effect of "bulging in the middle and shrinking at both ends," consistent with the conclusion in the paper by Dong et al. [39]. The bulging occurs at a position biased towards the loading end (45 mm away from the loading end).

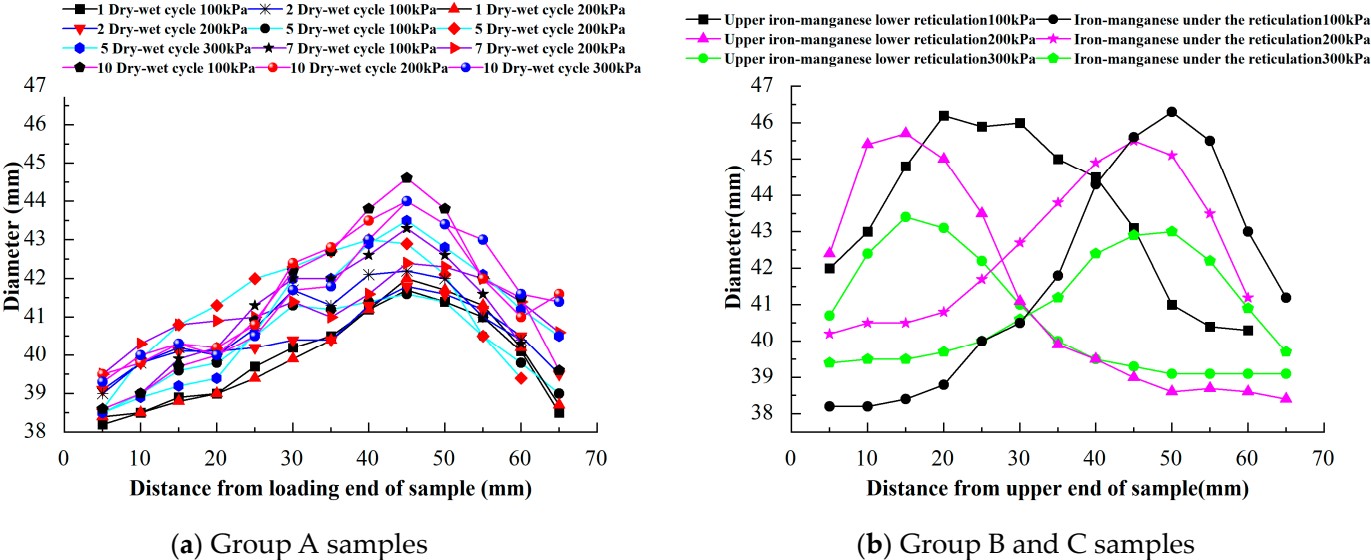

(**a**) Group A samples  (**b**) Group B and C samples

**Figure 5.** Lateral deformations of the paleo clay under different loading directions.

Figure 5b presents the influence of the confining pressure and loading position on radial specimen deformation. From the figure, it can be seen that under the same loading direction, regardless of the confining pressure, the specimen deformation shows the same trend. At the same time, it can be observed that when the iron-manganese clay layer is located at the loading end, the diameter change curve shows an upward and then downward process with the increase in confining pressure, and the peak diameter gradually decreases. This regular change can be attributed to the increase in confining pressure promoting the plastic deformation of the specimen [40]. However, after the confining pressures reaches the failure strength of the samples, the growth in radial deformation is limited, and with an increase in confining pressure, the specimen accelerates its attainment of yield strength. When the loading direction changes, the specimen exhibits significant differences in deformation, and the peak diameter is biased towards the loading end, consistent with the deformation law of the single-layer soil specimen mentioned earlier. When the confining pressure is same, for specimens with higher hardness at the loading end, their bulging deformation occurs closer to the loading end, and the deformation quickly reaches its peak, followed by a slow decline. Conversely, when the iron-manganese clay is at the unloading end, in other words, the loading end is relatively soft, the diameter of the deformation manifests a gradual increase towards its peak value, followed by a rapid subsequent decrease. Obviously, changing the loading end (loading direction) has little effect on the peak value of the diameter variation.

The area enclosed by the diameter change curve and the horizontal axis can reflect the volume change of the specimen. Therefore, by calculating the area enclosed under different confining pressures and loading directions, the compression of the specimen volume can be quantified, and the deformation law of the specimen can be analyzed. As indicated in Figure 5a, when the loading direction is constant, as the confining pressure increases from 100 kPa to 300 kPa, the area decreases, which represents a smaller specimen volume and smaller bulging deformation. The alteration in sample soil volume can be attributed to the presence of its load-bearing effect. This phenomenon instigates a profound reconfiguration within the structural composition of the soil, subsequently giving rise to pore compression within the soil matrix of the specimen. Consequently, this compels a densification of the constituent soil particles, culminating in their compaction. The alteration in the volume of the soil sample can be attributed to the influence of loading, which initiates a restructuring of the internal arrangement of the soil. This restructuring subsequently results in the compression of pores within the soil and the compaction of soil particles.

### 3.2. Effect of Loading Direction on the Strength of Heterogeneous Specimens

The relationship between the axial principal stress difference and the strain under various loading directions with three confining stresses is depicted in Figure 6. The peak point of the curve corresponds to its failure point, and in the absence of a peak value, the failure point can be determined by the principal stress difference at an axial strain of 13% [25].

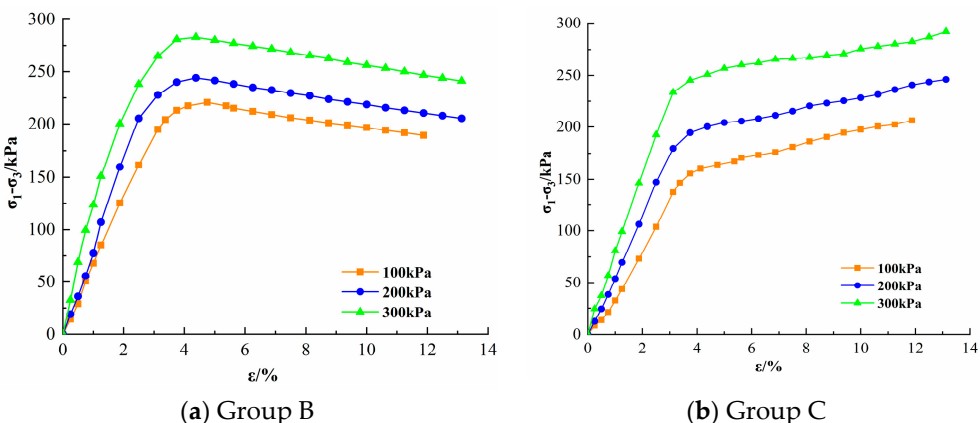

(**a**) Group B  (**b**) Group C

**Figure 6.** Stress–strain curves of the paleo clay under different loading directions.

Obviously, it can be observed that the stress–strain curve of the samples is considerably affected by the loading direction. For Group B samples at the beginning of the loading, with the increase in axial stress, the axial deformation becomes obvious, the main stress difference gradually increases linearly, and the stress–strain curve shows an overall rising trend. In the late stage of soil failure, the deviatoric stress of Group B is smaller than that of Group C under the condition of loading direction. This showed that during the initial stage of specimen failure, the structural integrity of the soil is damaged, resulting in a gradual decline in its resistance. Simultaneously, it becomes apparent that the paleo clay substrate is notably susceptible to the influence of the loading direction, consequently resulting in a stress–strain curve that exhibits a discernible propensity for strain softening. The stress–strain curve first exhibits an increase in peak stress, then a slight decrease in the various stresses (strain softening behavior). In contrast, the specimens within Group C demonstrate a distinctive strain hardening phenomenon, characterized by a gradual and sustained augmentation of stress levels subsequent to attaining the point of sample failure.

Correspondingly, elevated confining pressures directly correlate with heightened yield stress and axial strain magnitudes within the sample matrix. As the confining pressure incrementally escalates, the lateral displacements of the soil specimen undergo constraint

due to the applied confining pressure. The gradual compression of the sample engenders augmented inter-particle interlocking among the paleo clay constituents, thereby leading to the compaction of interstitial voids. Consequently, the magnitude of deviatoric stress requisite for the same degree of strain is proportionally amplified alongside the augmenting confining pressure levels. This was consistent with the previous observations that Group B exhibits a penetrating main rupture surface. The slope of the straight line in the early stages of the two groups of specimens is different, with Group B having a relatively larger slope, indicating that the specimens have a higher elastic modulus and exhibit low-strength material properties. The peak stress of Group B is relatively higher than that of Group C, indicating that pressure needs to be applied more vigorously to reach the failure strength of the samples.

According to the geotechnical test method, the Mohr–Coulomb failure envelope is depicted on the stress plane σ-τ, and the common tangent of each circle is drawn under different confining pressures. The obtained common tangent is the cohesion and the inclination is the internal friction angle (as shown in Figure 7). Based on the Mohr–Coulomb failure criterion, the shear strength parameters of the samples can be obtained by plotting the stress Mohr circle based on the experimental data (Table 2).

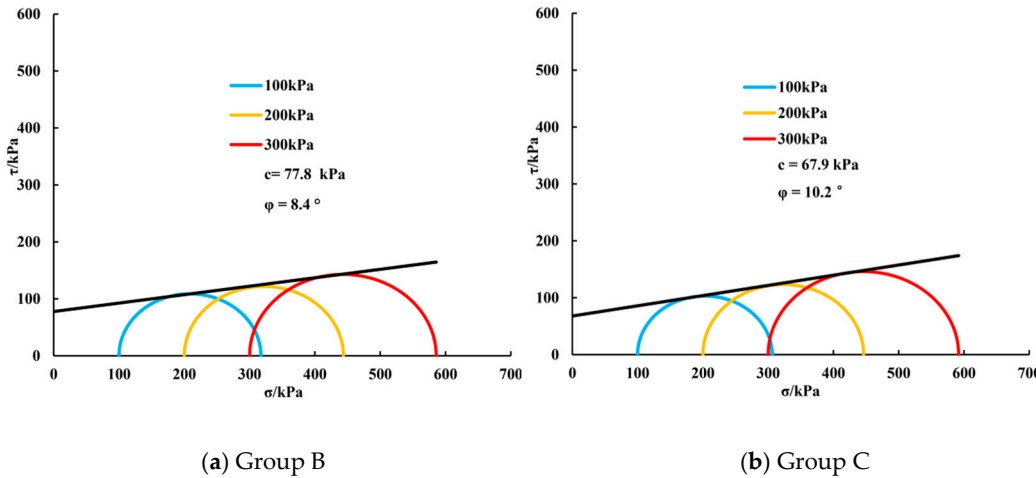

|  | (**a**) Group B |  | (**b**) Group C |
|--|--|--|--|

**Figure 7.** Mohr circle of stress under different loading directions.

**Table 2.** Shear strength parameters of heterogeneous paleo clay.

| Group | c (kPa) | φ (°) |
|---|---|---|
| B | 77.8 | 7.8 |
| C | 67.9 | 8.3 |

From Table 2, it is evident that the cohesive force (c) value of the samples in Group B surpasses that of samples in Group C by 14.58% when the loading direction is altered. However, the discrepancy in the friction angle (φ) is minimal, measuring only 0.5°. This observation highlights the significant influence of changing the loading direction on the strength of heterogeneous paleo clay specimens.

Based on the triaxial tests of reticulated clay specimens and the iron-manganese clay specimens of a single material, the cohesive force (c) value and internal friction angle of Group B are closer to those of the reticulated clay specimens, while the strength test results of Group C are closer to those of the iron-manganese clay specimens. The internal failure mode of the specimens changes due to the different stress processes. Therefore, the internal structural changes are distinct between the two groups of specimens. The specimens at the loading end exhibit smaller deformations, indicating that the loading end has higher strength. During the compression phase of a triaxial test on a soil sample, force

transmission is an essential process. Initially, the force is transmitted from the loading end and subsequently transferred to the soft soil sample. As the compression progresses over a specific duration, the force transmission persists. However, owing to the fixed condition of the lower end, further compression becomes unfeasible. Consequently, a reactive force is transmitted to the soft soil sample at the lower end.

### 3.3. Numerical Simulation

Although the processes of the triaxial tests were unified, the triaxial test results still showed great discreteness and poor regularity as a result of the randomicity of the clay samples. Generally, the triaxial test is expensive and requires a very long test cycle. However, numerical experimentation, an important auxiliary means in scientific research, provides an efficient research method that breaks through the limitations of conventional test equipment capacity and test conditions. Therefore, numerical experimentation was conducted on the homogeneous clay samples with three groups to access the effect of different loading directions on the failure sites of the specimens.

In the process of the numerical simulation, all of the simulated specimens were 4 cm in width and 11 cm in height. The left and right boundaries of the models were confined by 100 kPa confining pressure in the X-axis direction. Combined with laboratory tests and empirical parameters of nearby similar clay, model material parameters (Table 3) were determined. Then it was significant to assign parameters to the models. In order to study the influence of the loading direction on the failures of the specimens, the three simulated homogeneous samples were respectively tested under three different conditions. Among them, one model was loaded on the upper boundary with a constant speed of 0.5 mm/min while another end was completely constrained in the Y-axis direction. Similarly, one model was loaded on the lower boundary with the same constant speed under the same Y-axis direction constraint. And the third model was loaded on both the upper and lower boundaries. Moreover, concerning the stress field, only the gravity field was taken into account. The clay specimens were modeled using the Mohr–Coulomb model. Finally, the Shear strain cloud diagrams obtained from the numerical simulations are shown in Figure 8.

**Table 3.** Model material parameters.

| Name | $C$ (kPa) | $\varphi$ (°) | $\gamma$ (g/cm$^3$) | K (Pa) | G (Pa) | $\Psi$ (°) | $\sigma_t$ (Pa) |
|---|---|---|---|---|---|---|---|
| Paleo clay | 35.4 | 21.7 | 1.96 | $4 \times 10^7$ | $1.5 \times 10^7$ | 10 | 9000 |

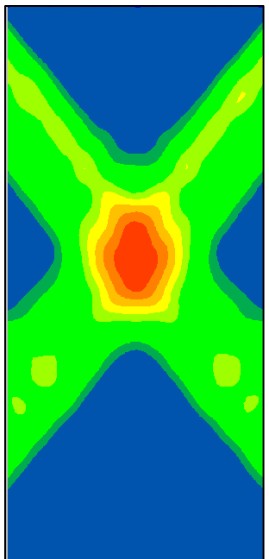 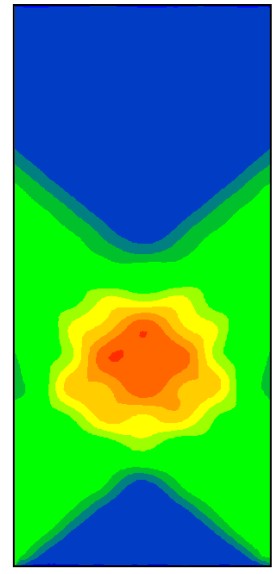 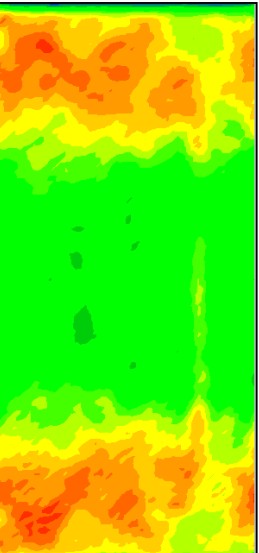

(**a**) Loading on the upper boundary   (**b**) Loading on the lower boundary   (**c**) Loading on both the upper and lower boundaries

**Figure 8.** Shear strain cloud diagrams.

The effect of different loading directions on the failure sites of the specimens is presented in Figure 8. From this figure, it can be clearly observed that the strain cloud diagrams are closely related to the loading direction. When both ends are loaded at the same time, the failures start from both ends and gradually develop into penetrating shear joints. In the process of loading on one of the upper and lower boundaries of the simulated specimens, X-type failures, which are regarded as an ideal failure model [41], form when the specimens are subjected to shear failures. When loading on the upper end, the failures tend to occur in the upper middle position of the samples. Vice versa, the main deformations can be observed in the lower middle parts when loading on the lower end. These results obtained from the numerical experimentation were completely consistent with the results of the triaxial compression tests. The phenomenon of the strain failure and deformation being closer to the loading end indicates that the failure of specimens is significantly affected by the loading direction, and the influence of the end effect decreases with the increase in the distance from the top surface. The stress–strain relationship and stress distribution of the middle area of the specimen are basically the same, and this part is less affected by the end effect.

## 4. Discussion

To reveal the influence mechanism of loading direction on the deformation and strength of the specimens, force analyses of failure processes combined with experimental phenomena and simulation results were performed. The loading model with the specimen loaded at both ends is presently still regarded as the traditional ideal. When applying this ideal loading model, when there is a force only loaded on one end of the specimen in the process of experiments, it is generally considered that the force is immediately transmitted to the other end, and finally the same effect as loading directly at both ends of the sample is achieved according to Newton's second law. However, the rationality of this equivalent loading method needs to be based on the premise that the specimen is a rigid material. In fact, for paleo clay samples, this is not suitable anymore. Therefore, a preliminary discussion of the influence mechanism of the loading direction on the deformation and strength of specimens is as follows.

The influence mechanism was further analyzed from the perspective of energy dissipation [42]. In this whole process, the energy shows an overall decreasing trend along the conduction path from the loading end to the other end of the specimen. It is well acknowledged that the compression process is not achieved overnight. The force is slowly transmitted through the medium of the specimen to the other end. Moreover, the loading end of the specimen is the first to produce deformation and consume energy. Under continuous compression, the elastic modulus of the material at the loading end is smaller, and the compressive strength is improved. At the same time, the material continuously absorbs part of the energy in this process so that the energy transmitted to the other end gradually decreases.

During the loading process, the force is transmitted to the other end of the sample from the loading end. At first, when the specimen is under confining pressure, the deformation of the specimen is dominated by axial deformation, and the lateral deformation of the specimen is weak. With continuous compression, it is clear that the volume of the specimen is reduced, the contact surface of the soil particles is increased, the porosity is reduced, and the density is increased. At the same time, the compressive strength of the soil at the loading end is improved under the confining pressure. Due to the diminishing disparity in strength between the opposing ends, the specimen undergoes progressive deformation, ultimately leading to its failure.

For non-uniform material samples loaded in different directions, when the high-strength section is compressed, the conduction velocity of energy is faster, which means that the strength gap between the two ends is enlarged. However, before the high-strength section reaches failure, the shear strength of the weak section is destroyed. Due to the large strength gap between the two ends, the weak section cannot extend the failure to the strong

section. Therefore, in this situation, compression deformation plays a leading role in the deformation of the two ends. This means that the loading direction has a significant effect on the deformation of heterogeneous specimens and even can change their failure modes.

For the analysis of the two loading directions, in the stress–strain relationship of group B, the first is the stage where stress increases with strain. The specimen will gradually deform as the strain increases, and the stress increases accordingly. After that, the stress reaches its peak, and some tiny cracks in the sample begin to expand, causing the strength of the sample to begin to decrease. The stress–strain curve of group C does not exhibit this phenomenon, but the stress continues to increase with strain. The above findings are of great significance for engineering applications. First, it can be used to analyze the response of the slope to deformation and mechanical effects in the direction of load, and secondly, help to predict the influence mechanism of the strength of multilayer soil in slopes.

## 5. Conclusions

In this study, triaxial compression experiments were conducted to examine the influence of the loading direction on the mechanical properties and deformation of paleo clay. The research integrates observations of specimen deformation and failure processes during testing with numerical simulation outcomes. Its objective was to investigate the impact of distinct loading directions on the deformation and strength of specimens consisting of two distinct layers of paleo clay, while delving into the underlying principles and mechanisms. Although the triaxial compression test is theoretically intended to apply loads at both ends, heterogeneous specimens exhibited a more pronounced alignment with loading outcomes at the unloading end, evident from comparisons between laboratory tests and numerical simulations. This indicates deviations in measuring parameters for heterogeneous materials using the triaxial compression test. The relevant conclusions obtained from this study are as follows:

(1) The loading direction plays an important role in the deformations of heterogeneous specimens and it even changes their failure modes. In the compression process, when the clay of the loading end has high hardness, overall shear failures are dominant and there will be a penetrating main fracture surface. When the hardness of the loading end is high, small bulging cracks are mainly formed in the low-strength section.

(2) Under the condition of unidirectional loading of heterogeneous material samples, the deformations at both ends of the samples are the smallest. The parts with significant deformations are concentrated at $40-50$ mm from the loading end.

(3) The loading direction has a significant effect on the stress–strain curves of heterogeneous material specimens. When the strength at the compressed end is low, the stress–strain curve exhibits the strain softening phenomenon with higher cohesion force. Inversely, strain hardening occurs when the strength of the loaded end is high.

(4) Under the numerical simulations, the manifestation of strain failure and deformation predominantly occurs in close proximity to the loading end. This observation underscores the pronounced influence of the loading direction on the failure behavior of samples. While examining the stress–strain relationship and stress distribution within the central regions of samples, remarkable consistency was noted. It is noteworthy that this central segment evinces diminished susceptibility to the influence of end effects.

(5) This study found that changing the loading direction induces a significant difference in the strength and deformation of multilayer paleo clay, which has guiding significance for landslide prevention and foundation engineering construction. Admittedly, this paper only tested a specific clay, which has certain limitations and more in-depth research is needed.

**Author Contributions:** Conceptualization, S.H. and Y.Y.; methodology, S.H.; software, S.H.; validation, Y.Y. and S.H.; formal analysis, W.C.; investigation, K.C. and R.X.; resources, W.C.; data curation, K.C. and Y.Y.; writing—original draft preparation, S.H.; writing—review and editing, S.H., S.H. and K.C.; visualization, S.H.; supervision, S.H. and Y.W.; project administration, Y.Y. and S.H.; funding acquisition, H.X. and S.H. All authors have read and agreed to the published version of the manuscript.

**Funding:** This research was funded by the Ph.D. Research Start-Up Fund of Hubei University of Technology, China, "Study on the evolution law of THMC characteristic parameters and constitutive model of solidified marine clay", grant number XJ2021000502; Natural Science Foundation of Hubei Province, China, "Spatial-temporal dynamic evolution mechanism of soil moisture in ecological protection slopes under rainfall conditions", grant number 2022CFB833; Young and Middle-Aged Talent Project of the Science and Technology Research Program of the Hubei Provincial Department of Education, China, "Experimental study on the mechanism of buried anti-slide piles and plants synergistic slope protection under rainfall conditions", grant number Q20275408; the Joint Funds of the National Nature Science Foundation of China (U22A20232); and the Project of Science and Technology Department of Shanxi Province, grant number of 202203021222428.

**Informed Consent Statement:** Not applicable.

**Data Availability Statement:** Data are contained within the article.

**Conflicts of Interest:** The authors declare no conflict of interest.

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
