# Peer review of "Effect of Loading Direction on Deformation and Strength of Heterogeneous Paleo Clay Samples"

_sustainability, doi:10.3390/su152215852_

Round 1

Reviewer 1 Report

Comments and Suggestions for Authors

This study investigated the macroscopic physical and mechanical properties of Paleo clay specimens during triaxial compression testing, aimed to elucidate the deformation mechanisms exhibit-ed by these specimens under varying loading directions at both the loading and unloading ends, and numerical simulation methods were carried out to simulate actual engineering scenarios.

This study is important for the real practice of engineering. By obtaining data from every angle, it will make a great contribution to the solutions of problems with real 3D analysis.

The following shortcomings have been identified in this manuscript;

In the abstract section, the most important result should be given and the contribution of the study should be given in general and explained in 1 sentence.

A picture of the clay used in the study should be added.

Overall, the work is nicely presented and well conveyed.

Author Response

Please view the reply in the file.

Reviewer 2 Report

Comments and Suggestions for Authors

As the lab tests are commonly known as reference tests for
field in situ tests, it is essential to analyze all
factors affecting results - also methodological.
Nevertheless, there are some issues which have to be discussed:

Some editorial issues.

Keywords "influence mechanism" is not precise.

Stiffness permeability - please use the more clear term

gramma issue
like "...stress state generated in the generation process, paleo clay is ..."

Five parameters are in Table 1. Basic physical and mechanical parameters of paleo clay - maybe something more?

What was the triaxial test type (CIU, CID, or UU)?

Give some references about your triaxial shearing test method.

What about pore water pressure during triaxial tests?

Are the confining pressures 100, 200, and 300 kPa total or effective values? What were cell pressure and pore water pressure?

Photos without scales.

"... the damaged stress circle ..." - do you have mean normal stresses at the moment of soil sample failure? -> The Mohr-Coulomb failure envelope. Use more common soil mechanics terms.

Why samples didn't have high/diameter ca. 2?

What were the strains of tested samples at the failure moments?

You used numerics to obtain a concurrent result with laboratory tests. The advantage of numerical simulations is the possibility of multiple test variants - could you present additional results to the three discussed in the publication?

Please write something more about your dry-wet cycles (figure 4).

A valuable phenomenological approach is presented in the paper.

Helpful literature that can be referenced:
1) isbn-10: 9780071363631
2) https://doi.org/10.1139/cgj-2013-0077
3) https://doi.org/10.1515/geo-2020-0291
4) doi:10.1520/D4767-11

Comments on the Quality of English Language

It's not bad, but still, the text needs native proofreading.

Author Response

Please view the reply in the file.

Reviewer 3 Report

Comments and Suggestions for Authors

General Comments:

The manuscript presents a study on the macroscopic physical and mechanical properties of Paleo clay specimens subjected to triaxial compression testing. The aim is to understand the deformation mechanisms under varying loading directions and simulate engineering scenarios. The research is valuable and contributes to the understanding of landslides and soil behavior. However, there are several issues and areas for improvement that need to be addressed before this manuscript can be considered for publication. The authors are strongly encouraged to address these points in a major revision of the manuscript.

Specific Comments:

Clarity of Introduction: The introduction provides a general context for the study but lacks a clear statement of the research objectives and the significance of the work. The authors should explicitly state what problem they are trying to address and why it is important. Additionally, it would be helpful to provide a brief overview of the methodology and key findings to give readers a sense of what to expect in the rest of the paper.

Methodology Description: The authors should provide a more comprehensive description of the triaxial compression testing procedure, including equipment used, sample preparation, and testing conditions. It would also be beneficial to explain the rationale behind choosing specific parameters and testing conditions.

Discussion and Interpretation: The discussion section is currently minimal and lacks in-depth analysis. The authors should provide a comprehensive interpretation of the results, discussing the implications of the observed deformation patterns and stress-strain relationships. Furthermore, the significance of these findings for engineering applications and landslide mitigation should be discussed.

Conclusions and Practical Implications: The conclusion section should be expanded to summarize the key findings of the study and their practical implications for engineering practices, especially in the context of landslides. The authors should highlight the contributions and limitations of their work.

References and Citations: The reference list appears to be incomplete, and some key references related to triaxial compression testing, clay behavior, and landslide studies are missing. The authors should ensure that the reference list is comprehensive and that all cited sources are properly referenced in the text.

Language and Style: The manuscript contains several grammatical and stylistic issues that need to be addressed. The authors should carefully proofread the manuscript to improve clarity and readability.

Author Response

Please view the reply in the file.

Reviewer 4 Report

Comments and Suggestions for Authors

This research studied the effect of different loading directions on mechanical properties and deformation of paleo clay. In this paper, a new method of define the loading direction by the position of two layers of paleo clay relative to the loading end, which has a good engineering application value. Specifically, the study performed some characterizations such as triaxial compression test at 100, 200, and 300kPa over two loading directions and found that changing the loading direction can cause the difference of deformation position as well as stress-strain curve. The overall structure of the article is very clear. Therefore, I suggest the paper is accepted after the modification. The specific review opinions are as follows:

 1. The conception of the article is novel, and the problem of the study is very worthy of attention.

  2. Figure. 3: The unit “KPa” in the graph should be modified to “kPa”.

3. Section 2.3: Why did the authors select three groups of samples for the test, and only group A selected samples with different dry-wet cycles?

4. In section Discussion: “The influence mechanism was analyzed further from the perspective of energy dissipation. In this whole process, the energy shows an overall decreasing trend along the conduction path from the loading end to the other end of the specimen.” This statement should include references.

5. Page 5, lines 165-168: “It is significant to carry out repeated tests, which namely that each (group of) three samples, correspondingly got from the three groups, should be repeated twice at least to improve the accuracy of the test results.” This sentence is not clear, please rewrite it.

Comments on the Quality of English Language

There are some language problems in some parts of the article. Please let the author address them carefully.

Author Response

Please view the reply in the file.

Round 2

Reviewer 2 Report

Comments and Suggestions for Authors

The article currently looks better. Minor editorial and linguistic corrections remain. Congratulations and good luck.

Comments on the Quality of English Language

Good but still worth reading carefully for best understanding.

Author Response

Please view the reply in the file.

Reviewer 3 Report

Comments and Suggestions for Authors

Accept

Author Response

Thank you for your comments.